# Autoimmune Diseases and Plasma Cells Dyscrasias: Pathogenetic, Molecular and Prognostic Correlations

**DOI:** 10.3390/diagnostics14111135

**Published:** 2024-05-29

**Authors:** Laura Giordano, Rossella Cacciola, Paola Barone, Veronica Vecchio, Maria Elisa Nasso, Maria Eugenia Alvaro, Sebastiano Gangemi, Emma Cacciola, Alessandro Allegra

**Affiliations:** 1Hematology Unit, Department of Human Pathology in Adulthood and Childhood “Gaetano Barresi”, University of Messina, Via Consolare Valeria, 98125 Messina, Italy; laura.giordano@polime.it (L.G.); paola.barone@polime.it (P.B.); vero.vecchio99@gmail.com (V.V.); maria.nasso@studenti.unime.it (M.E.N.); m.eugenia983@gmail.com (M.E.A.); 2Hemostasis/Hematology Unit, Department of Experimental and Clinical Medicine, University of Catania, 95123 Catania, Italy; rcacciol@unict.it; 3School and Operative Unit of Allergy and Clinical Immunology, Department and Experimental Medicine, University of Messina, 98125 Messina, Italy; gangemis@unime.it; 4Department of Medical and Surgical Sciences and Advanced Technologies “G.F. Ingrassia”, University of Catania, 95123 Catania, Italy; ecacciol@unict.it

**Keywords:** autoimmune disease, multiple myeloma, MGUS, monoclonal gammopathies, systemic lupus erythematosus, psoriasis, rheumatoid arthritis

## Abstract

Multiple myeloma and monoclonal gammopathy of undetermined significance are plasma cell dyscrasias characterized by monoclonal proliferation of pathological plasma cells with uncontrolled production of immunoglobulins. Autoimmune pathologies are conditions in which T and B lymphocytes develop a tendency to activate towards self-antigens in the absence of exogenous triggers. The aim of our review is to show the possible correlations between the two pathological aspects. Molecular studies have shown how different cytokines that either cause inflammation or control the immune system play a part in the growth of immunotolerance conditions that make it easier for the development of neoplastic malignancies. Uncontrolled immune activation resulting in chronic inflammation is also known to be at the basis of the evolution toward neoplastic pathologies, as well as multiple myeloma. Another point is the impact that myeloma-specific therapies have on the course of concomitant autoimmune diseases. Indeed, cases have been observed of patients suffering from multiple myeloma treated with daratumumab and bortezomib who also benefited from their autoimmune condition or patients under treatment with immunomodulators in which there has been an arising or worsening of autoimmunity conditions. The role of bone marrow transplantation in the course of concomitant autoimmune diseases remains under analysis.

## 1. Introduction

Multiple myeloma (MM) is the second most prevalent hematologic malignancy, found in the spectrum of plasma cell dyscrasias [1]. Neoplastic transformation of a plasma cell clone causes hyperproduction of identical immunoglobulins (Figure 1), resulting in monoclonal gammopathy of undetermined significance (MGUS) and, due to the occurrence of additional mutations, leading to multiple myeloma. It is classically a pathology of the elderly, although cases have been observed in the young population, usually with a poor prognosis [2,3,4,5].

MGUS is defined as an increase in monoclonal immunoglobulin (Ig) in blood or urine of less than 3 g/dL, clonal plasma cells less than 10% in the bone marrow and the absence of any clinical signs. MGUS is present in 3% of individuals aged 50 and above, and its occurrence becomes more common as age advances [6,7,8].

While the exact cause of MGUS and MM is not well understood, there is evidence suggesting that immunological dysfunction or prolonged immune system activation may be crucial factors in the development of both disorders. Monoclonal immunoglobulin is commonly found in chronic inflammatory illnesses, including chronic infection and autoimmune disorders [9,10,11,12]. The influence of these two entities on immunological homeostasis is particularly significant when taking into account the increased vulnerability to severe infectious complications, even during times free from myelotoxic therapies. This is due to a change in the humoral immunity’s effector arm. The progressive accumulation of dysfunctional plasma cells in the bone marrow results in a direct suppression of B lymphocytopoiesis and non-clonal immunoglobulin production. In fact, a characteristic marker of multiple myeloma is the decrease in, and occasionally the complete absence of, physiological immunoglobulins. This immune paralysis results in a decrease in the patient’s capacity to develop an effective primary defense against infections and an incapacity to create a strong secondary defense [13,14,15].

An autoimmune disease (AD) is a clinical illness that occurs when T cells and/or B cells are activated in the absence of an ongoing infection or any other recognized cause. The basis for autoimmune diseases (ADs) lies in the failure to distinguish between self and non-self and the disruption of immunological tolerance. During these pathological situations, T cells cause damage to tissues by directly destroying target cells, attracting inflammatory cells and releasing various cytokines. Autoantibodies (autoAbs) may trigger tissue injury by forming immune complexes, causing cytolysis or phagocytosis of target cells and disrupting cellular function (Figure 2).

Central and peripheral immune tolerance controls the activity of T cells and B cells. Nevertheless, certain autoreactive T and B cells migrate to the outer regions, where they remain dormant until an external stimulus disrupts the tolerance and stimulates the innate and adaptive immune cells in an individual with genetic susceptibility [16,17,18].

Chronic inflammation is known to have a role in the development of hematological malignancies and other forms of cancer. Different inflammatory pathways are implicated in B cell survival. They are mediated by interleukin 6 (IL-6), interleukin 13 (IL-13) and tumor necrosis factor (TNF)-α. Toll-like receptor (TLR) and its ligands stimulate the growth of B cells, while B-cell activating factor (BAFF) and the subsequent activation of nuclear factor κ-B (NF-κB) are associated with B cell neoplasias.

In recent years, different studies have suggested a connection between plasma cell dyscrasias and ADs. It has been hypothesized that prolonged exposure to antigen stimulation could cause plasma cells’ malignant transformation by leading to pro-oncogenic mutations in rapidly dividing cells [19,20,21,22].

Increased activity of B cells is a characteristic feature of these autoimmune disorders, and there is evidence to suggest that immune-related conditions may be linked to an increased risk of MGUS and MM. A significant incidence of MGUS has been seen in a cohort of patients with immunological diseases or persistent infections. Nevertheless, the precise mechanism connecting benign inflammatory conditions and the development of genetically clonal cells is still uncertain [9]. An increased risk for MGUS and MM has been found in families with a positive history for ADs [23]. In particular, a correlation between a personal history of giant cell arteritis (GCA) and polymyalgia rheumatica and a higher risk of MGUS and MM has been under study [24]. The exact causes are unknown, but a possible role of the long-term immunological activation has been suggested. Furthermore, the connection between ADs and MGUS seems stronger than that between ADs and MM [25,26].

McShane et al. discovered that a 42% higher incidence of MGUS was associated with any autoimmune condition. Patients with detectable autoantibodies and organ involvement showed a higher risk, while those with non-detectable autoantibodies showed a less significant elevated risk [25].

Eight families were subjected to genetic analysis, which demonstrated that hyperphosphorylated paratarg-7 was transmitted as a dominant character and that carriers had an almost eight-fold greater risk of multiple myeloma and IgA or IgG MGUS. 

In addition, greater correlations were identified in patients with MGUS rather than with MM (42% vs. 13%) [27,28,29].

## 2. Physiopathology of Autoimmune Diseases and Multiple Myeloma

### 2.1. Hematological Conditions

Anemia is one of the CRAB criteria and it is frequently present at the moment of MM diagnosis. It has a complex base regarding the inflammatory environment caused by MM itself, the invasion of the bone marrow by plasma cells and the erythropoietin deficiency following renal impairment. Nonetheless, other causes of anemia have been reported: pure red cell aplasia, pernicious anemia and autoimmune hemolytic anemia (AIHA) [19,30].

ADs and plasma cell diseases seem to be closely related to a number of blood crasis changes. The potential connections between certain illnesses, including AIHA, idiopathic thrombocytopenia purpura (ITP), autoimmune neutropenia (AIN), acquired hemophilia (AHA) and coagulation problems with monoclonal gammopathies, will be assessed in this section. 

While some research indicates that AIHA is extremely infrequently related with MM, others assert that 4% of individuals with MM also have AIHA. A recent study revealed the presence of antibodies against erythrocyte surface antigens in 10.6% of the population under analysis, suggesting that the production of autoantibodies could be facilitated by concurrent immunological abnormalities in patients affected by MM [31,32,33]. 

ITP has also been reported in MM patients, although rarely, with a similar pathogenic cause [34,35].

AIN is defined by an absolute neutrophil count (ANC) of less than 1500 cells/mL. It could have secondary causes such as rheumatological diseases, ITP, AIHA and cancer. In MM, neutropenia is typically the consequence of treatment. Up to 35% of patients develop neutropenia (grades 3 and 4) brought on by lenalidomide [19,30]. 

Some aspects point to MM as the most likely underlying cause of AIN: neutrophil antibodies, the absence of other possible causes, the timing of AIN’s diagnosis and its resolution with the control of MM. Despite the fact that the precise cause of AIN in MM is still unknown, MM and immunological dysregulation are related. Immunological dysregulation in MM may cause AIN, or AIN may cause immunological dysregulation in MM [36].

AHA is a rare autoimmune disease responsible for an increased bleeding tendency due to the presence of autoantibodies initiating the activity of coagulation factor VIII. Approximately half of AHA cases are associated with an underlying condition, such as cancer, ADs or pregnancy complications. In the remaining cases, AHA’s cause is still unknown.

AHA can arise as a result of hematological malignancies or be associated with them. Among these cancers, lymphoproliferative diseases are the most prevalent. Only 14% of AHA cases associated with a hematological malignancy are specifically related to plasma cell neoplasms (PCNs) [36].

Coagulation system abnormalities pose a substantial problem in patients with plasma cell disorders. Approximately 6–7% of patients with MGUS and 10% of individuals with primary amyloidosis experience venous thromboembolisms. Immunomodulatory drugs used to treat MM, such as thalidomide, lenalidomide or pomalidomide, greatly increase the risk of thrombotic events. However, patients with plasma cell abnormalities do develop bleeding tendencies due to impaired platelet production, altered platelet function caused by paraproteins and vascular injury resulting from hyperviscosity syndrome. Some cases have demonstrated that therapy for MM decreases the occurrence of coagulopathy by lowering the paraproteins in the blood.

These findings imply that paraproteins are a major factor in the aberrant coagulation observed in MM patients, working as neutralizing antibodies for certain coagulation factors [37,38,39,40].

### 2.2. Rheumatological Conditions

According to the literature, there are several pathologies of rheumatological interest associated with MM. The dysregulation of the immune response resulting from the state of chronic inflammation and the consequent alteration of immunological surveillance up to the release of cytokines by pathological plasma cells seem to be common hypotheses [19]. 

In patients affected by systemic lupus erythematosus (SLE), tolerance mechanisms such as elimination of the plasma cells producing autoantibodies and editing their receptor to create non-self-reacting immunoglobulins are overridden by B cell hyperactivity. The main targets of these pathogenic plasma cells are nuclear components, including DNA and histone. Both humans with SLE and lupus-prone mice exhibit a tendency towards excessive activity of B cells [41]. 

There is evidence to suggest that individuals with autoimmune conditions have an increased risk of developing cancer. According to a recent study, even in the absence of cytotoxic therapies, individuals with SLE had a higher chance of having a tumor. Among the neoplasms associated with SLE, lymphomas—particularly non-Hodgkin’s lymphomas—are the most often reported forms. Plasma cell dyscrasias, including macroglobulinemia and MGUS, have also been linked to SLE (11% of patients) [19], even though it is uncommon for SLE and MM to coexist [42].

In patients affected by SLE, it has been suggested that B cell hyperactivity facilitates the escape of atypical B cell clones from the regulatory systems. Mice with spontaneous lupus-like symptoms had a high frequency of plasma cell dyscrasias [43]. There may be a direct link between MM and chronic inflammatory diseases, as evidenced by the finding that both conditions may be associated with an increase in bone marrow plasma cell counts. An alternate theory is that SLE patients’ compromised immune surveillance, which makes them more susceptible to cancer overall, encourages the emergence of MM [44].

A similar pathogenetic mechanism was hypothesized in patients affected by Sjögren’s syndrome (SS), a chronic autoimmune disease of unknown etiology which is characterized by a progressive infiltration of lymphocytes and plasma cells in the salivary and lacrimal glands [45]. Patients with primary SS have an increased relative chance of having B cell lymphoproliferative disorders compared to the normal population. It has been estimated that one-third of patients with SS develop a B cell lymphoma. Polyclonal B cell activation is the central immunoregulatory aberration in SS, although T cells represent the majority of the lymphoid cells invading these glands. Hyperglobulinemia and the development of many autoantibodies against both organ-specific and non-organ-specific autoantigens are manifestations of this activity. Treatment for MM in a patient who was also affected by SS resulted in a reduction in SS symptoms, such as xerostomia and xerophthalmia. While the involvement of a hyperimmune reaction in the development of lymphoma in SS is recognized, the potential link between chronic inflammation in SS and the onset of MM remains to be determined [46]. 

Furthermore, the idea of impaired immune surveillance as a cause of both immune and neoplastic components could explain the association between MM and another AD, primary biliary cirrhosis (PBC) [47,48]. PBC is an autoimmune disease causing a persistent state of hepatic inflammation, leading to the gradual loss of small interlobular bile ducts. This progresses to the onset of cholestasis, liver damage and eventually the development of fibrosis and cirrhosis [49]. It has been linked to higher levels of immunoglobulins, the presence of circulating immune complexes and a greater likelihood of developing extrahepatic malignancies. There have been four reported instances of a connection between multiple myeloma and PBC. In two patients who had targeted treatment for the hematological condition, there was a decrease in cholestasis indices and immunoglobulins, indicating a potential correlation between the two disorders [50,51].

Following the case of neurological involvement, two conditions appear to have a significant clinical role: chronic inflammatory demyelinating polyradiculoneuropathy (CIDP) and giant cell arteritis (GCA). CIDP is a peripheral neuropathy characterized by both distal and proximal sensory impairments, often related to lymphoproliferative diseases [52,53,54]. Multiple myeloma has been associated with many forms of neuropathy, including axonal demyelinating. Neuropathy in MM typically manifests as a mild, symmetrical sensorimotor deficit, characterized by greater sensory impairment than motor impairment, and with a preference for affecting the distal parts of the body over the proximal parts. Idiopathic CIDP cannot be distinguished from CIDP associated with monoclonal gammopathies. At the time of diagnosis, clinical signs of neuropathy are observed in 33% of individuals with multiple myeloma. More than 50% of multiple myeloma cases exhibit detectable subclinical neuropathy. Certain cases display an elevated level of antibodies against neural antigens, despite the absence of clinical neuropathy [55].

GCA, a chronic inflammation of blood vessels responsible for the irroration of scalp and head, has been correlated to MM, suggesting that they might share a mutual inflammatory mechanism. Specifically, GCA may be associated with an immunological response that occurs as a result of cytokine release in MM, or it may be caused by amyloid deposits that are subsequent to MM [56].

It is not possible to report an estimate of the incidence of some conditions such as PBC, CIDP and GCA as we referred to studies with scant case series, but they were carefully considered to support our association between ADs and MM.

## 3. A Closer Look: Molecular Aspects

Focusing our attention on the molecular basis, several soluble and biological factors play a key role in the pathogenesis of MM and ADs. B lymphocytes are found in an inflammatory microenvironment, supported by pathways involving IL-6, IL-10 and tumor necrosis factor alpha [57]. 

Among the pro-inflammatory cytokines, the leading role was attributed to IL-6. Inflammation has been linked to both monoclonal B cell neoplasia and polyclonal B cell activation. Individuals who have a history of chronic inflammations also tend to have polyclonal hypergammaglobulinemia and often develop lymphomas or plasma cell neoplasias. When combined, these findings indicate that the development of autoantibodies and polyclonal B cells is closely linked to the chronic-inflammation-induced IL-6 production [58]. 

Regarding a deeper prospect, different proteins have been studied. Tumor necrosis factor (TNF) family proteins participate in the control of vital cell functions such as differentiation, proliferation, survival and cell death. Changes in the expression of TNF family members are frequently linked to pathological states such as cancer and autoimmune diseases. First reported in 1998, the TNF-like ligand APRIL (a proliferation-inducing ligand) received its name from its ability to promote tumor cell proliferation in vitro. Originally discovered in hematopoietic cells under normal circumstances, APRIL is overexpressed in several tumor tissues [59].

From a biological perspective, regulatory B lymphocytes appear to play an essential part. They are in charge of controlling the immune response by undermining T cells’ defenses and helping them transform into regulatory T cells (Tregs). The inflammatory aspect of the MM environment stimulates angiogenesis, adaptive immunity and the growth and survival of malignant cells, all of which have a profound effect on the regulatory cells’ ability to operate. It is commonly known that immunological abnormalities in MM are seen not only in B cells but also in natural killer (NK), T cells and dendritic cells (DCs) [60] (Table 1).

### 3.1. IL-6

Interleukin 6 (IL-6) is a versatile cytokine responsible for the development of B cells as well as playing a role in the immune response, hematopoiesis and inflammation. The hypothesis that IL-6 is implicated in autoimmunity was advanced while studying patients affected by cardiac myxoma. In fact, the cardiac myxoma cells were found to produce IL-6 and the patients exhibited autoimmune symptoms [61].

Elevated levels were also discovered in the synovial fluid of patients diagnosed with rheumatoid arthritis (RA), and it is essential for the induction of autoimmune disorders and autoimmunity in experimental settings. These facts indicate that IL-6 has a significant role in the B cell abnormalities linked to the inflammatory process. In line with this concept, inflammation has been linked to both widespread activation of B cells and the development of abnormal growth of a single clone of B cells [62]. Individuals who have pre-existing chronic inflammation often exhibit an excessive production of multiple types of antibodies (polyclonal hypergammaglobulinemia) and are prone to developing abnormal growths of plasma cells (PCs) or lymphoma. Collectively, these findings indicate that the synthesis of IL-6 that results from chronic inflammation is strongly associated with the activation of multiple B cells and the formation of autoantibodies. The underlying cause of SLE is believed to be the improper activation of B cells, which are responsible for producing antibodies. This abnormal activation is assumed to be caused by a dysregulated IL-6/IL-6R pathway in B cells. In the absence of T cells, B cells derived from individuals with SLE exhibit autonomous production of substantial quantities of IgGs, including IgG antibodies targeting single-stranded DNA (ssDNA). Activated B cells respond to IL-6 by generating IgGs and IgG anti-ssDNA antibodies. Significant IL-6 production and B cell expression of IL-6R have been linked to SLE and a number of other autoimmune disorders. While regular T and B cells do not express IL-6, B cells associated with SLE produce larger quantities of IL-6. IL-6 transgenic mice exhibit a significant increase in the production of polyclonal plasma cells, together with the generation of autoantibodies and the development of glomerulonephritis characterized by the proliferation of mesangial cells. The concept that dysregulated IL-6 gene expression can initiate polyclonal plasmacytosis and subsequently lead to the development of a malignant monoclonal plasmacytoma is well supported by the available data [63,64,65]. IL-6-deficient mice do not develop plasmacytoma, which aligns with this concept [58].

### 3.2. Differentiation of Plasma Cells

PCs are B cells that have been activated by antigen and undergone a transformation that enables them to produce substantial amounts of neutralizing antibodies. They are quiescent, terminally differentiated cells which remain in the niches of the bone marrow (BM) for a few days or possibly years. Two forms of PCs can be distinguished: short-lived and long-lived plasma cells. When B cells encounter antigens with the assistance of T cells through the process of CD40 ligation (known as T-dependent responses), the B cells generate prolonged responses [66].

Activated B cells in the germinal center (GC), known as centroblasts, proliferate rapidly and undergo processes of isotype switching and somatic hypermutation. These events alter the hypervariable regions of the Ig genes, hence modifying the specificity of the B cell receptor (BCR) [67,68].

The cells resulting from this process are centrocytes, which subsequently go through an affinity maturation phase: their survival is dictated by their affinity to the BCR. They can then leave the germination center and become memory B cells or long-lived PCs. Multiple cell surface indicators are downregulated in long-lived PCs [69]. Surface CD45, B220, MHC class II, CD19 and, most significantly, the BCR are all lost in this process. The ability of long-lived PCs to make antigen-specific antibodies over prolonged periods of time is, of course, their most remarkable feature. Long-lived PCs endure for at least a year, whereas short-lived PCs could barely last a few days [70]. The sequential expression and silencing of transcription factors that regulate B cell destiny can be utilized for tracking the progress of B cell development. B lymphocyte-induced maturation protein 1 (BLIMP-1) is essential to the PC transformation process, as several investigations have demonstrated. Bcl-6 and B cell lineage-specific activator (BSAP, encoded by PAX5) are both repressed by BLIMP-1 expression and are necessary for the early phases of B cell maturation. Numerous additional genes are also repressed by BLIMP-1, such as MHC class II transactivator (CIITA), which is necessary for MHC class II expression, and activation-induced cytidine deaminase (AID), which is necessary for SHM and isotype class switching. PC formation is lost when BLIMP-1 expression is suppressed. Therefore, the type of B cell activation and the anatomical site of residency have a major impact on the final quality and size of a PC population. The interplay between early developmental phases and environmental management will play a key role in the establishment of a PC population. PCs’ unique microenvironment has been linked to factors that affect cellular survival and lifetime and, consequently, their ability to generate Ig over extended periods of time. Chemokines and their receptors regulate migration to particular microenvironmental niches. The interaction between PC and BM stroma also appears to affect plasma cells’ lifespan [71].

Dysregulation of B cells’ maturation and differentiation is the base of autoimmune diseases and certain hematological cancers. The first step is the immortalization of PCs, resembling long-lived plasma cells. Expression of BLIMP-1, oct-2, PU.1 and Spi-B was detected in myeloma cells; they are essential for the differentiation of plasma cells and enhancing the production of immunoglobulins. In particular, PU.1 controls the expression of many integrins necessary for stem cell homing and engraftment onto bone marrow stroma [72,73]. The presence of PU.1 in pathological plasma cells can indicate their level of development and help explain the mechanism by which MM is mainly located in the bone marrow [74].

### 3.3. APRIL: A Proliferation Inducing Ligand

The TNF family of ligands and receptors plays a crucial role in controlling several cellular processes, including activation, survival and cell death. A proliferation inducing ligand (APRIL) and B-cell activating factor belonging to the TNF family (BAFF) are two strongly associated constituents of the TNF ligand superfamily [75]. APRIL was first defined in 1998 and given its name due to its ability to promote the growth of cancer cells in both laboratory settings and living organisms. APRIL is manifested by a diverse range of immune cell subsets. The production of APRIL by monocytes, macrophages and dendritic cells relies on the presence of specific cytokines in the environment, such as interferon (IFN) gamma and IFN alpha. The expression of APRIL receptors on B lymphocytes varies depending on their stage of maturity and activity. APRIL and BAFF, previously recognized as B cell modulators under normal settings, were hypothesized to play a role in supporting the survival of chronic lymphocytic leukemia (CLL) cells. They were found in both MM cell lines and in the bone marrow of MM patients, and elevated levels of circulating APRIL and BAFF are also observed in the blood of MM patients. In the bone marrow of multiple myeloma patients, MM cells, CD14+ cells and osteoclasts all express both ligands. In particular, osteoclasts appear to be the major producers of APRIL/BAFF. Osteolytic bone disease affects 70–80% of MM patients. This condition is mostly caused by increased osteoclast activation and interactions between MM cells and the bone marrow microenvironment. Plasma cells co-cultured with osteoclasts show an enhanced viability. Thus, it can be suggested that APRIL and BAFF, generated from osteoclasts, have a role in the proliferation of MM cells. This further suggests that APRIL and BAFF participate in B cell malignancies [59].

### 3.4. The Role of Innate Lymphoid Cells in ADs and MM

In recent years, the anti-tumor properties previously attributed to tumor-infiltrating leukocytes have been rediscussed, and it is now common knowledge how they are indeed responsible for tumor progression and the development of metastases. It is precisely in this scenario that innate lymphoid cells (ILCs) make their appearance: they are lymphocyte-like cells of the innate immune system with a significant role in inflammatory and neoplastic processes. In a neoplastic environment, ILCs evade immune surveillance mechanisms and release pro-neoplastic factors [76]. They have a lymphoid morphology and lack specific antigen receptors and phenotypic markers of myeloid cells and dendritic cells (DCs) [77].

They are distinguished in three categories on the basis of phenotypic features: ILC Group 1, including NK and ILC1 cells; Group 2 with ILC2 and Group 3 including lymphoid tissue inducer (LTi) and ILC3 cells [78]. Group 1 NK cells are fundamental in cancer surveillance. Previous studies demonstrated that patients with MGUS and newly diagnosed MM have higher levels of CD56+ CD3− NK cells in their blood and bone marrow and that these patients have a worse prognosis [79]. 

ILCs are also an important source in the production of cytokines. ILC1s secrete IFN-γ, which is thought to have a protective effect against cancer. IFN-γ in fact inhibits the proliferative activity of MM cells by suppressing IL-6, which is a critical growth factor for MM. Another important cytokine is IL-10. In a malignant environment, it acquires a negative prognostic value: IL-10 inhibits the release of pro-inflammatory IFN-γ and TNF-α and enhances the emergence of NK-resistant tumor phenotypes. ILC1s maintained the capacity to release lineage-specific cytokines (IFN-γ) in individuals with MGUS, whereas this ability was notably diminished in those with asymptomatic MM [80].

ILC2 is regarded as a cell subtype with pro-tumor features. It in fact secretes cytokines that promote tumor growth and inhibit anti-tumor immunity in the microenvironment. ILC2s in patients with MGUS were shown to secrete IL-13 [81].

The role of ILC3s in MM and MGUS is not entirely clear due to the limited research available. However, their involvement in the development and progression can be hypothesized: ILC3s have exhibited pro-tumor activity in several cancers by releasing cytokines such as IL-17, IL-22 and IL-23, which are implicated in the development of inflammatory illnesses [82].

Moreover, ILCs seem to also have a role in the pathogenesis of ADs. Different studies have discovered evidence of ILCs’ alterations in human psoriasis. Investigators utilized immunofluorescence in situ staining to demonstrate the distribution of ILCs in psoriatic skin, particularly at the epidermis and in close proximity to T cells, indicating a direct contact between these cell types [83].

Research indicates an increase in ILC3s and ILC1s but a decrease in ILC2s in psoriatic skin lesions, suggesting that ILC1s and ILC3s may contribute to the disease, whereas ILC2s could have a protective effect in psoriasis development [84,85]. The pathogenetic role of ILC3s in Ps may be attributed to the production of crucial pro-inflammatory cytokines such as IL-17 and IL-22 [83].

### 3.5. The Role of Multi-Omics Analysis in MGUS and ADs

Numerous biomarkers of human diseases can be found and quantified using novel omics approaches such as transcriptome, proteome and metabolome profiling. Thus, this integrative multi-omics analysis can aid in our methodical understanding of the molecular pathways that underlie disease and biological function. Our understanding of the pathophysiology of autoimmune diseases will grow with the integration of transcriptome, proteome and metabolome studies data [86]. 

For instance, omics techniques are widely used in SLE research. These techniques help the identification of beneficial compounds and signal transduction pathways. Scientists have found through transcriptomics that patients with SLE have abnormal mRNA expression of biomarkers mediated by immune response and inflammatory signals, cell proliferation and differentiation signals and type I interferon signaling [87]. 

Proteomics approaches have revealed abnormal autoantibodies, chemokines, complement system, cytokines and novel protein biomarkers such as VCAM-1 and C-reactive protein in SLE patients [88]. Researchers have found, thanks to metabolomics, that energy-producing, purine nucleotide, oxidative stress and fat-related metabolic signals are aberrant in SLE [88].

Furthermore, a study identified IFN-driven changes in the composition and phenotype of T and NK cells that are consistent with systemic immune activation. This was achieved by using a targeted single-cell multi-omics approach that combined protein and mRNA quantification to generate a high-resolution map of the T lymphocyte and natural killer (NK) cell populations in blood from SLE patients [89]. The study of monoclonal gammopathies is also pertinent to omics approaches. After being developed, a transcriptome-based response predictor model demonstrated encouraging predictive accuracy in MM patients undergoing first-line therapy [90]. By employing multi-omics techniques, it will be possible to uncover novel pathogenetic mechanisms and perhaps diverge from current therapeutic approaches, as well as identify new shared pathways between autoimmune illnesses and monoclonal gammopathies. Future research and clinical trials will illustrate the value of various multi-omics expression analysis platform components as biomarker finding tools for the diagnosis, prognosis and therapy of autoimmune entities [91]

## 4. Impact of Autoimmune Disease on Multiple Myeloma Prognosis

The connection between ADs and plasma cell dyscrasias has now been established. However, what the burden is on patients who have both conditions must be questioned. Some investigations have tried to find an answer, discovering an increased risk of concurrent MGUS but not MM in patients with previous ADs [25]. Individuals having a personal history of giant cell arteritis, polymyalgia rheumatica or autoimmune hemolytic anemia were found to have an elevated risk of multiple myeloma [92]. The precise biological mechanisms underlying these observations remain unclear; however, it has been suggested that they may arise from a shared genetic predisposition or persistent antigen stimulation, which in turn triggers the formation of plasma cell dyscrasias [25]. Focusing on the risk of progression from MGUS to MM, there has been proven to be a lower risk of developing MM, particularly in patients without detectible autoantibodies in blood [93].

A population-based study revealed a 17% lower risk of progression, also indicating that while MGUS is more prevalent in patients with ADs, it tends to have a less severe evolution [94].

Other analyses confirmed similar deductions; thus, it has been hypothesized that MGUS in patients also affected by ADs has a different biological base. Even patients affected by smoldering multiple myeloma (SMM) and concomitant ADs appear to have better prognostic features. However, the risk remains: a higher prevalence of MGUS in this population leads to a higher probability to develop MM, even if with a slower course [92]. 

Potential underlying causes of the lower risk of progression may be that MGUS in a setting with chronic low-level antigen stimulation is biologically less likely to undergo the genetic events that trigger progression. Additionally, the use of immunosuppressive therapies for the autoimmune conditions may delay or prevent progression [95].

Nevertheless, when considering prognosis and survival, individuals with MM and a prior history of autoimmune diseases exhibited an elevated mortality rate. Possible explanations encompass the existence of common genetic factors, the likelihood that individuals with a history of autoimmunity encounter more severe cases of MM or a combination of comorbidity in each patient [96,97].

## 5. Psoriasis, Multiple Myeloma and Impaired Immune Tolerance

Psoriasis (Ps), a papulosquamous skin disease, was originally thought to be a disorder primarily of epidermal keratinocytes but is now recognized as one of the most common immune-mediated disorders. An immune dysregulation seems to be the cause of both Ps and MM, resulting in an impaired balance of effector and regulatory cells, such as B regulatory lymphocytes (Bregs) [98,99,100]. 

Specific attention was placed on the progenitor population of the regulatory cells, CD19+CD24hiCD38hi Bregs lymphocytes, which is also implicated in the production of cytokines with regulatory activity on the immune system, such as IL-10, and therefore responsible for the autoimmune process, which is advantageous in the case of Ps but also of tumorigenesis, which appears to be facilitated by such immunosuppressive mechanisms [101].

### Regulatory B Cells

ADs are caused by a breakdown of immunological “tolerance” and a failure to distinguish self from non-self. Under these pathogenic conditions, T cells cause damage to tissues through cytolysis of target cells by enlisting the help of inflammatory cells and producing various cytokines. Additionally, autoantibodies (autoAbs) can form immune complexes, induce phagocytosis or cytolysis of target cells and interfere with cellular function. T cells and B cells are under the influence of both central and peripheral immunological tolerance. Nonetheless, a portion of autoreactive T and B cells go to the periphery, where they stay inactive until an external stimulus breaks tolerance and activates innate and adaptive immune cells in a genetically susceptible individual. Even some Tregs may be induced to generate pro-inflammatory cytokines as a result of inflammation [16,102].

Regulatory B cells (Bregs) are essential to the pathophysiology of neoplasms and autoimmune disorders. They are responsible for producing immunological tolerance by generating inhibitory molecules such as programmed cell death ligand 1 (PDL1) and suppressive cytokines such as interleukin-10 (IL-10) and interleukin-35 (IL-35). Bregs’ IL-10 production level does, in fact, reflect their regulating role [103,104]. 

They cause effector T lymphocytes to die by binding to CD40 on the surface of Bregs and to its ligand CD40L on T cells, which reduces the response to the autoantigen. 

Autoimmune and autoinflammatory diseases may be exacerbated by an imbalance in these immune effector actions. However, the overexpression of regulatory cells might play in favor of the onset of tumors [105,106].

The microenvironment is crucial in the development of MM; inflammation promotes angiogenesis and the growth and survival of cancerous cells. Other immune cell abnormalities have also been noted. The development of the illness may be linked to myeloid-derived suppressor cells, which create a repressive environment in the bone marrow [107]. At this stage, MM develops into relapsed/refractory MM (RRMM) and immunotherapy becomes ineffective [108,109,110].

Even though B cells minimize T effector cell responses during chronic inflammation, which may be advantageous in psoriasis (Ps), malignant cells thrive in this environment. The mechanism appears to be self-sustaining: the pathogenic plasma cells enhance the survival of Bregs by reducing their programmed cell death, and the resulting IL-10 facilitates immunosuppression [111]. The percentage of regulatory B lymphocytes appears to be lower in Ps-affected patients, but CD19+CD24hiCD38hi Bregs, ancestors of Bregs, are higher in percentage. Following Ps immunosuppressive therapy, this equilibrium appears to shift in favor of a higher concentration of B regulatory cells [60,112].

In MM, the production of IL-10 in CD19+CD24hiCD38hi Bregs appears to be correlated to the severity of the disease, and it was in fact found to be elevated in patients with ISS stage III.

Indeed, IL-10 has an immunosuppressive role in MM, which aids in the disease’s development. It may also promote B cells’ differentiation into pathogenic plasma cells, which is associated with a worse prognosis [113].

In one study, the percentages of CD19+CD24hiCD38hi Bregs were considerably lower after daratumumab therapy. Remarkably, the proportions of CD19+CD24hiCD38hi Bregs dropped very quickly following the first treatment cycle. Nonetheless, there was no discernible difference in RRMM patients’ plasma levels of IL-10 before and after daratumumab therapy, suggesting other potential sources of IL-10. However, the study’s findings showed that daratumumab therapy for RRMM resulted in an instantaneous reduction in CD19+CD24hiCD38hi Bregs [101,114].

## 6. Role of Multiple Myeloma Therapy in Autoimmune Diseases

In the therapeutic scenario of MM, there are different approaches built on the biological characteristics of the disease and patient characteristics such as comorbidity, age and performance status. To date, daratumumab, bortezomib, dexamethasone and immunomodulators such as thalidomide and lenalidomide used in combination are the first line of treatment for newly diagnosed transplant-eligible patients [115].

Regarding the treatment of ADs, a challenge arises: the pathological target is represented by a peculiar type of plasma cells, found in MM and also in ADs. They are in fact long-lived plasma cells, resident in specialized niches of the marrow and therefore not easily attacked by the therapeutic means available to us. This has generated interest in the search for molecules capable of acting against this type of plasma cell, and numerous ideas come from the therapy against multiple myeloma [116,117].

### 6.1. Daratumumab

Daratumumab is a human immunoglobulin G kappa monoclonal antibody targeting CD38. Daratumumab-based regimens improved the depth of response, including MDR negativity, in newly diagnosed cases of multiple myeloma as well as in relapsed/refractory MM [118,119,120,121]. The production of autoantibodies is carried out by a specific type of plasma cell: long-lived, non-dividing plasma cells. They provide a therapeutic challenge since they are found in specific niches inside the bone marrow and are shown to resist immunosuppressive treatments.

A recent study on patients with SLE found that daratumumab successfully decreased the number of plasma cells in patients’ peripheral blood and also suppressed the activity of CD38 on the remaining plasma cells [122].

A subgroup of lupus patients were recently found to have an expanded CD38highCD8+ T cell type with decreased cytotoxic capacity in association with an increased propensity to infections, despite the fact that the circulating plasmablasts in those patients had CD38 expression levels comparable to those of healthy individuals [107]. It was discovered that the usage of daratumumab was linked to excellent clinical and serologic responses in two patients with refractory lupus and in patients affected by SS [123].

Given that anti-CD38 therapy with daratumumab is effective in MM and that long-lived plasma cells exhibit elevated levels of CD38, there is a possibility that this treatment will lead to the therapeutically meaningful elimination of pathogenic long-lived plasma cells. Indeed, a considerable (about 60%) decrease in pathogenic anti-dsDNA antibodies was related to daratumumab. Improvements in many clinical characteristics, including pericarditis, autoimmune hemolytic anemia, lupus nephritis and mucocutaneous symptoms in SLE patients, were attributed to the use of daratumumab [124].

It has been suggested that daratumumab may have additional effects such as blocking T cells co-expressing CD38 which are responsible for lupus nephritis and depleting plasmacytoid dendritic cells, also expressing elevated levels of CD38, hence reducing interferon type I activity. The majority of patients with SLE display an increased expression of type I interferon (IFN)-regulated genes, lacking proper negative feedback mechanisms and thus becoming one of the driving forces behind the disease [125].

Daratumumab also reduces CD19 B cells by 50%, depleting the autoreactive B cell components, and it plays a role in the transcriptional profile of CD4 and CD8 cells, downregulating the gene transcripts linked to the repeated antigenic stimulation [126].

### 6.2. Bortezomib

Degradation of redundant and misfolded proteins, along with a range of regulatory proteins, is made possible by the 26S proteasome, a proteolytic complex found in the cytosol and nucleus of eukaryotic cells. The 26S proteasome complex is thought to be responsible for the processing of more than 80% of all cellular proteins, including nuclear factor kappa B (NF B) and inhibitor protein kappa B (I B), both of which are crucial for the onset of cancer and inflammation. There have been numerous autoimmune and inflammatory illnesses linked to the proteasomal system. Serum samples from patients with SLE, myositis, rheumatoid arthritis and Sjögren’s syndrome showed a correlation between the levels of circulating proteasomes and the severity of their conditions [127].

The approved medication for MM, bortezomib, selectively and reversibly inhibits the 26S proteasome. Its ability to reduce inflammation is linked to its suppression of NFB. In a human skin graft, bortezomib decreased the psoriatic lesions’ size, thickness, keratinocyte proliferation and leukocyte infiltration. These findings offered convincing evidence for investigating bortezomib’s potential application in the management of psoriasis [128].

In MM, apoptosis occurs when the neoplastic plasma cells’ proteasome catalytic sites are inhibited. By affecting dendritic cells and T and B cells and blocking the NF-kB pathway, proteasome inhibition suppresses the immune system by lowering the generation of pro-inflammatory cytokines. Short-lived and long-lived plasma cells (SLPCs, LLPCs) are impacted by BTZ, which also lowers autoAbs, essential for the development of autoimmune disorders [16,129,130].

### 6.3. Lenalidomide

An Italian retrospective study reported an elevated occurrence of autoimmune diseases subsequent to lenalidomide therapy. Autoimmune cytopenias were the most common manifestation of ADs. Additionally, cases of polymyositis, Graves’ illness, optic neuritis and vasculitis were noted, and thyroid problems have been reported to occur in up to 6% of patients receiving lenalidomide therapy in the past [131].

The evolution toward an autoimmune dysfunction might suggest the existence of an underlying immunological imbalance that lenalidomide, with its powerful immunostimulatory qualities, can turn on in favor of its autoreactive component to an overt AD [132,133].

### 6.4. Effect of Transplantation for Multiple Myeloma on Autoimmune Disease

It has been demonstrated that when high-dose chemotherapy is paired with autologous stem cell transplantation (HSCT) rather than just conventional chemotherapy alone, patients with myeloma have improved overall survival, event-free survival and even quality of life [115]. There are numerous cases of long-term remission of concurrent autoimmune diseases following autologous or allogeneic stem cell transplantation in individuals with hematological malignancies [134]. Due to these findings, there is growing interest in the combination of autologous stem cells and high-dose chemotherapy for the treatment of different forms of autoimmune diseases. To improve their management, a rigorous preparative immunoablative regimen is thought to be able to eliminate the patient’s immune cells, including autoreactive lymphocyte clones. Nonetheless, there have also been reports of autoimmune disorder recurrence following autologous stem cell transplantation in some hematologic malignancy coexisting with an autoimmune disease.

Coexisting cases of RA have frequently temporarily improved with autologous SCT for hematological malignancy; however, the RA typically relapses shortly after transplantation. The reason why patients with myeloma have a greater rate of post-transplantation autoimmune condition recurrence is still unknown [135]. 

Shaikh H. et al. monitored a patient with concurrent ankylosing spondylitis (AS) and MM who had a long-lasting remission after HSCT. One explanation could be that the patient’s melphalan conditioning altered the autoimmune T cell clone, which led to the rheumatological process’s remission. Given the potential that AS in these cases may be a paraneoplastic condition, it is possible that the MM treatment caused the concomitant remission of the autoimmune disease [136].

## 7. Conclusions and Future Prospectives

MM and ADs share complex and heterogeneous pathogenetic bases. We attempted to bring to light the possible mechanisms that explain their connection, concluding that a correlation between the two pathologies is possible: it is expressed in the context of the inflammatory medullary microenvironment, the production of long-lived PCs capable of producing autoantibodies and becoming immortal and the set of cytokines and proliferation/anti-apoptotic factors that control their survival [9,11,44,137,138]

The possibility of treatments aimed at controlling both pathologies remains to be discussed. The potential of bone marrow transplantation in eradicating them is weighted by collateral factors that are not always predictable such as the selection of clones of autoreactive lymphocyte due to the delayed recovery of CD4+ cells [134,135]. 

Current therapies available for MM such as daratumumab and bortezomib have proven useful in controlling ADs by acting against long-lived PCs located in specialized niches of the bone marrow that are often difficult to reach for common AD treatments, leaving hope for a common therapeutic approach and further strengthening the hypothesis that the two pathologies have a close connection, with pathogenetic bases capable of influencing each other [44,95].

The newest frontier in the treatment of MM is represented by bispecific antibodies (bsAbs). They are proteins against two targets: the tumor cell and an effector cell, typically a T cell. The bsAb activates the T cell, which forms a cytolytic synapse with the tumor cells, causing the release of cytotoxic molecules responsible for tumor apoptosis [115]. 

Studies have demonstrated the possibility of using bsAbs also in the context of autoimmune diseases [139,140]. Currently, two bsAbs, blinatumomab (CD19 × CD3) and emicizumab (Factor IX/Factor X), have been authorized in the treatment of oncological and hematological malignancies [141]. BsAbs for autoimmune diseases are still in the early phases of research, even though significant developments have been made in the treatment of rheumatoid arthritis, SLE, asthma, Ps and other autoimmune diseases. All these results provide a further contribution to the knowledge of these pathologies and could constitute a substrate for the development of new treatment strategies [141].

## Figures and Tables

**Figure 1 diagnostics-14-01135-f001:**
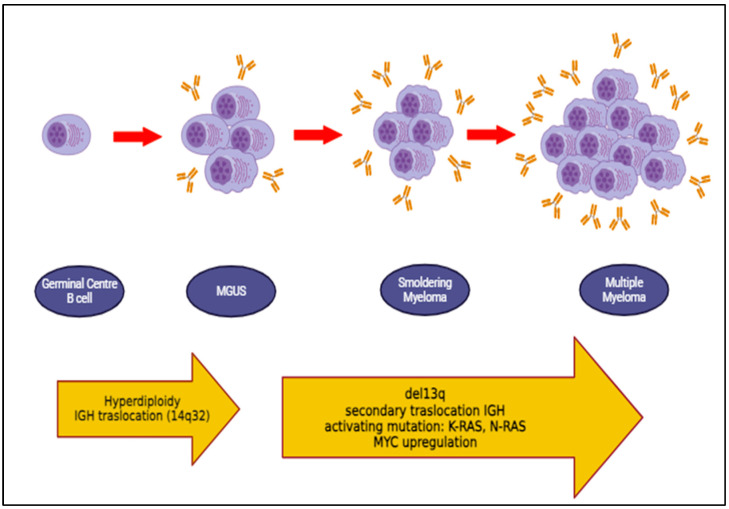
Pathogenetic mechanisms of myeloma. Created with BioRender.com (accessed on 28 February 2024).

**Figure 2 diagnostics-14-01135-f002:**
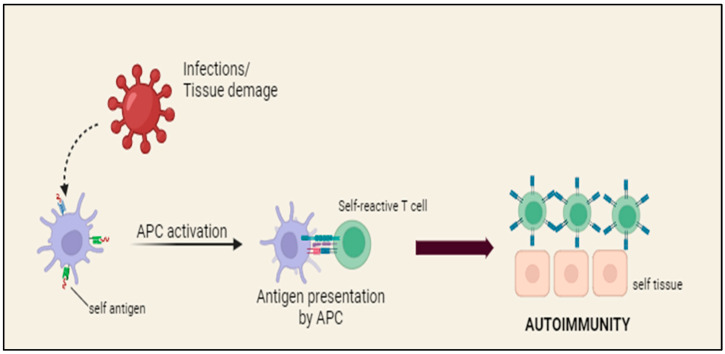
Pathogenetic mechanism of autoimmunity.

**Table 1 diagnostics-14-01135-t001:** Cytokines and other factors in ADs and MM.

Cytokines and Other Factors	Role in ADs	Role in MM
Interleukin 6	Activation of B cells → abnormal growth → autoantibodies production	Plasmablasts growth, maturations of plasmablasts in pathological PCs
Interleukins 13 and 4	Anti-apoptotic effect, IgE production	B cells proliferation
APRIL and BAFF	Chronic inflammation (activation of keratinocytes in Ps and joint erosion in RA)	Survival of MM cells Inhibition of apoptosis Ig class-switch recombination
Interleukin 10	Immunosuppression	Progression of MM B cells differentiation in MM PCs
Interleukin 35	B- and Tregs expansion	B cells differentiation in MM PCs

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
