# Peer review of "Autoimmune Diseases and Plasma Cells Dyscrasias: Pathogenetic, Molecular and Prognostic Correlations"

_diagnostics, 2024, doi:10.3390/diagnostics14111135_

Round 1

Reviewer 1 Report

Comments and Suggestions for Authors

There are wealth of literature on this subject describing interesting findings based on gene expression analyses to identify unique predictive autoimmune signatures supporting the need for a personalised approach to treatment of AD patients.

The advancement in disease diagnostics is rapidly shifting from isolated medicine to more personalized medicine The rapid expansion and transformation of the modern diagnostic tools can be attributed to developments in proteomic and genomic technologies (functional genomics). Hence this topics need to be covered in the review with respect to AD.

1.     It is not clear whether the cited references were mainly from information from clinical studies findings or studies of basic and applied research were also included.

2.     This review paper will not be completed without a section focusing particularly some prominent aspects regarding potential clinical utility of different components of  Multi Omics expression analysis platforms as biomarker discovery tool for prognosis, diagnosis, and treatment of these autoimmune entities.

Author Response

There are wealth of literature on this subject describing interesting findings based on gene expression analyses to identify unique predictive autoimmune signatures supporting the need for a personalised approach to treatment of AD patients.

The advancement in disease diagnostics is rapidly shifting from isolated medicine to more personalized medicine The rapid expansion and transformation of the modern diagnostic tools can be attributed to developments in proteomic and genomic technologies (functional genomics). Hence this topics need to be covered in the review with respect to AD.

  1. It is not clear whether the cited references were mainly from information from clinical studies findings or studies of basic and applied research were also included.

Of course, the referee's observation is relevant. Actually, by combining both clinical and fundamental research data, we have attempted to provide a comprehensive overview of the data that are currently available in the literature on the subject.

  1. This review paper will not be completed without a section focusing particularly some prominent aspects regarding potential clinical utility of different components of  Multi Omics expression analysis platforms as biomarker discovery tool for prognosis, diagnosis, and treatment of these autoimmune entities.

We revised the content and included the paragraph 3.5 on  page 10 of 21 lines 411-442 discussing the value of multi-omics expression analysis systems for studying monoclonal gammopathies and autoimmune disorders and including  the references 86-91.

Reviewer 2 Report

Comments and Suggestions for Authors

The present article reviewed scientific literature on the association between plasma cell disorders and autoimmune diseases.

The contents are complete and sound. The authors covered epidemiological, pathogenetic and clinical issues. The impact of stem cell transplant, one of the most relevant issues, has been discussed.

References are adequate.

There is only one minor point That I recommend to address. Overall, MM is associated with reduce immune function and reduced immunoglobulin levels. Despite the attempts, the author couldn't, in my opinion, discuss this discrepancy enough. Please make this part of the discussion more robust

Author Response

The present article reviewed scientific literature on the association between plasma cell disorders and autoimmune diseases.

The contents are complete and sound. The authors covered epidemiological, pathogenetic and clinical issues. The impact of stem cell transplant, one of the most relevant issues, has been discussed.

References are adequate.

There is only one minor point That I recommend to address. Overall, MM is associated with reduce immune function and reduced immunoglobulin levels. Despite the attempts, the author couldn't, in my opinion, discuss this discrepancy enough. Please make this part of the discussion more robust.

As suggested, we added an insight on the subject, explaning the correlation as a pathogenetic aspect on page 2 of 21 lines 56-64 including the references 13-15.

Reviewer 3 Report

Comments and Suggestions for Authors

Dear authors,

you've tried to review a hige number of autoimmune diseases in connection or in contradiction with plasma cells dyscrasias. However, it looks a bit controversial and/or excessive. You've given a lot of information but haven't support it with any results. There are no data on incidences of autoimmune diseases among, for example, multiple myeloma patients and healthy controls, or vise versa. No clear conclusion had been made. I think you should try to make your article more clear and precise. 

Comments on the Quality of English Language

Main comment concerns acronyms - they are rarely introduced at the site of first citation, usually they occur on second or third citation. Some of the acronyms had been introduced twice, a few were given without explanation. 

Then, some references lack commas, parenthesis, spaces.

Author Response

Dear authors,

you've tried to review a hige number of autoimmune diseases in connection or in contradiction with plasma cells dyscrasias. However, it looks a bit controversial and/or excessive. You've given a lot of information but haven't support it with any results. There are no data on incidences of autoimmune diseases among, for example, multiple myeloma patients and healthy controls, or vise versa. No clear conclusion had been made. I think you should try to make your article more clear and precise. 

We revised the concerned paragraph, including the missing data where available to paragraph 2.2 on page 5 of 21 lines 181-182 and lines 196-197 and including the sentence “For some conditions such as PBC, CIDP and GCA the estimate of the incidence was not possible to make as we referred to studies with scant case series, but which were carefully considered to support our association between ADs and MM” to paragraph 2.2 on page 6 of 21 lines 237-239.

Main comment concerns acronyms - they are rarely introduced at the site of first citation, usually they occur on second or third citation. Some of the acronyms had been introduced twice, a few were given without explanation. 

Then, some references lack commas, parenthesis, spaces.

The issue has been addressed in the review file.

Round 2

Reviewer 3 Report

Comments and Suggestions for Authors

Dear authors, I'm satisfied with the corrections you've made.